# AutoBio: A Simulation and Benchmark for Robotic Automation in Digital Biology Laboratory

**Zhiqian Lan**[1,2*]  **Yuxuan Jiang**[1,2*]  **Ruiqi Wang**[3]  **Xuanbing Xie**[2]  **Rongkui Zhang**[3]
**Yicheng Zhu**[4]  **Peihang Li**[1]  **Tianshuo Yang**[1]  **Tianxing Chen**[1]  **Haoyu Gao**[3]
**Xiaokang Yang**[4]  **Xuelong Li**[2]  **Hongyuan Zhang**[1]  **Yao Mu**[4†]  **Ping Luo**[1,5†]
[1]The University of Hong Kong   [2]China Telecom, TeleAI   [3]Tsinghua University
[4]Shanghai Jiaotong University   [5]HKU Shanghai Intelligent Computing Center

## Abstract

Vision-language-action (VLA) models have shown promise as generalist robotic policies by jointly leveraging visual, linguistic, and proprioceptive modalities to generate action trajectories. While recent benchmarks have advanced VLA research in domestic tasks, professional science-oriented domains remain under-explored. We introduce AutoBio, a simulation framework and benchmark designed to evaluate robotic automation in biology laboratory environments—an application domain that combines structured protocols with demanding precision and multimodal interaction. AutoBio extends existing simulation capabilities through a pipeline for digitizing real-world laboratory instruments, specialized physics plugins for mechanisms ubiquitous in laboratory workflows, and a rendering stack that support dynamic instrument interfaces and transparent materials through physically based rendering. Our benchmark comprises biologically grounded tasks spanning three difficulty levels, enabling standardized evaluation of language-guided robotic manipulation in experimental protocols. We provide infrastructure for demonstration generation and seamless integration with VLA models. Baseline evaluations with SOTA VLA models reveal significant gaps in precision manipulation, visual reasoning, and instruction following in scientific workflows. By releasing AutoBio, we aim to catalyze research on generalist robotic systems for complex, high-precision, and multimodal professional environments.

## 1 Introduction

Vision-language-action (VLA) model architectures jointly leverage vision, language, and proprioception modalities to generate action trajectories in an end-to-end manner. By aligning capabilities more closely with how humans perceive and interact with the world, these models hold promise as a pathway toward generalist robotic policies. Recent VLA models (Brohan et al., 2023; Kim et al., 2024; Ghosh et al., 2024; Liu et al., 2025; Black et al., 2024; 2025) have demonstrated impressive results in real-world scenarios, including table bussing, clothing folding, and household manipulation. However, current benchmarks remain largely confined to domestic settings, leaving a critical gap in evaluating VLA models for professional, science-oriented scenarios.

Biology laboratories represent a uniquely promising yet challenging environment for robotic automation. Experiments in these environments follow clear, rigorous protocols (Schilling et al., 2008), making them well-suited for language-guided robots to interpret and execute. Additionally, the repetitive and time-consuming nature of many laboratory tasks presents opportunities for automation to reduce researcher workload and enhance efficiency. Existing solutions, such as automated sample preparation platforms (May, 2016), often prioritize throughput at the expense of

---

*Equal contribution.
†Corresponding authors. Email: pluo@cs.hku.hk, muyao@sjtu.edu.cn

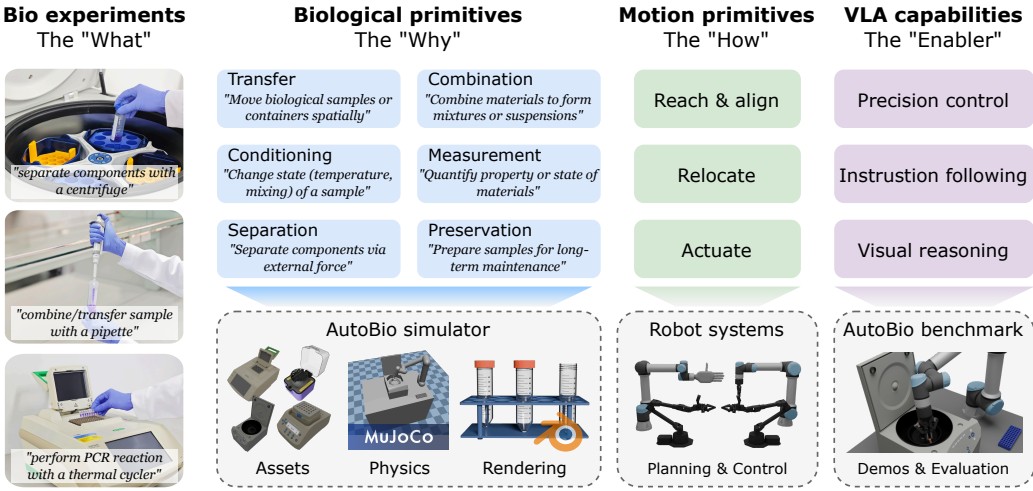

Figure 1: AutoBio framework. AutoBio decomposes complex experiments into fundamental biological primitives. These are then implemented via robotic motion primitives within a specialized simulation environment. AutoBio simulator features instrument digitization pipeline, custom physics plugins for lab mechanisms, and rendering stack supporting dynamic interfaces and transparent materials. This enables creation of biologically grounded benchmark tasks to evaluate VLA models on precision control, instruction following, and visual reasoning capabilities in scientific workflows.

flexibility, whereas robotic systems could achieve human-level adaptability. Nevertheless, biological experiments impose distinct challenges on robotic intelligence: Long-horizon workflows require perception and interaction with diverse interfaces (e.g., digital displays, control panels, and articulated mechanisms). Precision-dependent tasks like slot alignment are ubiquitous, while transparent liquids and containers complicate visual reasoning. These characteristics make biology laboratories an ideal benchmark for evaluating VLA capabilities in language grounding, visual understanding, and high-precision manipulation.

To realize this potential, however, it is critical to first address the challenges of simulating biology laboratories—a task that demands advanced asset modeling, physics simulation, and rendering capabilities. Robotic simulations typically use physics engines such as MuJoCo (Todorov et al., 2012), Bullet (Coumans & Bai, 2016–2021), and PhysX to provide a foundation for physics-based interaction. Building upon these engines, each simulator offers its own set of features tailored to specific focus areas. For instance, robosuite (Zhu et al., 2020) and MuJoCo Playground (Zakka et al., 2025) primarily target reinforcement learning (RL) (Sutton et al., 1998) algorithms through standardized tasks involving pick-and-place operations, articulated object manipulation, and locomotion. RoboTwin (Mu et al., 2025) emphasizes dual-arm coordination and proposes an automated pipeline for data synthesis. These simulations generally address out-of-the-box contact-based rigid body interactions, and lack specialized capabilities required for biological experimentation.

In the AutoBio simulator, we extend asset modeling, physics simulation, and rendering capabilities to streamline the entire pipeline of biological experiment simulation, specifically targeting the biological primitives outlined in Figure 1. In biological experiments, many operations rely on specialized instruments such as centrifuges, thermal cyclers, and mixers. We develop a workflow to transform real-world instruments into manipulable assets within AutoBio through 3D Gaussian Splatting (Kerbl et al., 2023), CAD refinement, and texture baking. These transformed instruments are then paired with self-contained logic to faithfully replicate their real-world characteristics. Regarding physics simulation, while the physical world operates under universal laws, physics engines typically provide only generalized implementations, leaving finer-grained interactions to be extended by users. To address this, we develop a suite of MuJoCo plugins supporting interactions prevalent in biological experiments—including thread mechanisms, detent mechanisms, eccentric mechanisms, and quasi-static liquid computation—that are rarely addressed in existing simulators. Finally, rendering fidelity proves crucial as vision serves as the primary input modality for VLA models. The prevalence of transparent materials in laboratory environments poses particular challenges, as conventional blend-mode rasterization engines handle transparency poorly. We integrate our simulation

Table 1: Comparison of AutoBio with related robotic manipulation simulators / benchmarks

| Benchmark (Simulator) | Target domain | #Tasks | Interactive instrument | Threaded object | Fluid | Reactive display | Render backend | Demo synthesis | VLA model train & eval |
|---|---|---|---|---|---|---|---|---|---|
| Meta-world | - | 50 | × | × | × | × | MuJoCo | ✓ | × |
| Robosuite | - | 9 | × | × | × | × | Isaac Sim | × | × |
| Factory | Factory | 8 | × | ✓ | × | × | Isaac Gym | × | × |
| Maniskill 2 | - | 20 | × | × | × | × | SAPIEN | ✓ | × |
| Robotwin | - | 14 | × | × | × | × | SAPIEN | ✓ | ✓ |
| Libero | Home | 130 | × | × | × | × | Isaac Sim | ✓ | × |
| Chemistry3D | Chemistry | 5 | × | × | ✓ | × | Isaac Sim | × | × |
| AutoBio (ours) | Biology | 16 | ✓ | ✓ | ✓ | ✓ | Blender | ✓ | ✓ |

with Blender's Physically Based Rendering (PBR) pipeline to achieve visually accurate transparency effects for containers and liquids. In addition, we implement dynamic texture rendering for instrument displays, enabling interactive UI manipulation—a critical feature for tasks involving digital interfaces.

By combining the above-mentioned simulation features, we propose the AutoBio benchmark, which distills robotic manipulation primitives into practical biological experiment tasks. We provide comprehensive infrastructure to facilitate trajectory synthesis, demonstration generation, and standardized data interfaces for VLA model integration. Our benchmark comprises tasks across three difficulty levels, where we evaluate open-source SOTA VLA models, including $\pi_0$ (Black et al., 2024), $\pi_{0.5}$ (Black et al., 2025) and RDT (Liu et al., 2025). This evaluation reveals critical limitations in current approaches and suggests potential improvements in model architecture and training methodologies.

Our key contributions summarize as follows: (1) A simulator designed for biology labs, featuring instrument digitization, physics plugins (thread/detent mechanisms, quasi-static liquids), and PBR rendering for transparency and reactive displays. (2) A benchmark with biologically grounded tasks, enabling standardized evaluation of robotic automation in lab protocols, with support for trajectory synthesis and VLA integration. (3) Systematic evaluation of VLA models in science-oriented settings, revealing critical gaps in precision manipulation, instruction following and visual reasoning.

## 2 RELATED WORKS

**Vision-Language-Action Model.** Recent generalist policies integrate vision, language, and control for broad task generalization. RT-2 (Brohan et al., 2023) leverages web-scale vision-language data for robotic control, while OpenVLA (Kim et al., 2024)—trained on 1M real-world demos—surpasses larger closed models on various manipulation tasks. RDT (Liu et al., 2025) extends diffusion transformers to bimanual manipulation, enabling zero-shot generalization to novel objects and scenes. $\pi_0$ (Black et al., 2024) (and its successor $\pi_{0.5}$ (Black et al., 2025)) combines VLMs with flow matching for smooth, cross-embodiment action generation. However, the benchmark tasks typically involve coarse manipulation (picking, placing, folding) rather than precise lab procedures. Long-horizon, high-precision tasks and interacting with digital interfaces are rarely evaluated. AutoBio is intended to fill this gap by focusing on lab-specific objects and instruments, explicitly testing fine-grained vision-language-action reasoning not covered by prior work.

**Robotic Manipulation Benchmark.** Existing benchmarks like ManiSkill (Gu et al., 2023), Meta-World (Yu et al., 2019), robosuite (Zhu et al., 2020), Libero (Liu et al., 2023), ARMBench (Mitash et al., 2023), and Factory (Narang et al., 2022) focus on rigid-object manipulation (e.g., pick/place, assembly) in home, warehouse or factory settings. While Chemistry3D (Li et al., 2024b) introduces science-oriented tasks for chemical experiments, and BEHAVIOR-1K (Li et al., 2024a) covers deformable settings, most lack support for fluids, digital interfaces, or lab-specific precision. AutoBio addresses this gap by simulating biological workflows with fluid handling, transparent materials, and interactive instrument UIs—challenges absent in prior benchmarks. A feature comparison with related benchmarks can be found in Table 1.

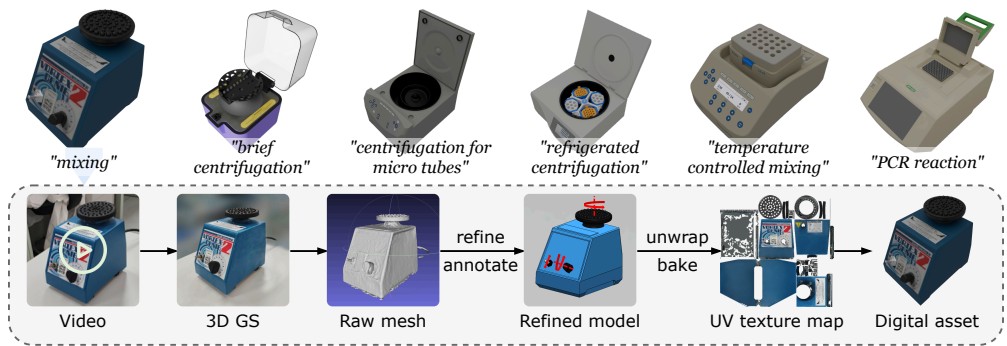

Figure 2: Digitized instruments for fundamental biological experiment operations, with an example taken from vortex mixer demonstrating the proposed workflow for digitizing real-world instruments.

**Robotic Automation in Laboratory.** A parallel research thread builds specialized "self-driving" labs for science. Szymanski et al. (Szymanski et al., 2023) describe A-Lab that fully automates solid-state materials synthesis with fixed hardware pipelines tailored to inorganic chemistry. In their system, robotic arms dose powders, handle furnaces, and transfer samples between instruments while ML algorithms propose recipes. Similarly, other efforts have automated enzyme engineering or pharmaceutical screens (Holland & Davies, 2020; Rapp et al., 2024; Tom et al., 2024). However, these platforms typically prioritize throughput: each piece of equipment is specialized for a specific protocol, and human oversight is still needed for setup or maintenance (Holland & Davies, 2020). AutoBio targets the complementary goal of flexible, human-like manipulation in the lab. Instead of crafting one-purpose machines, AutoBio's framework is designed to move toward handling novel protocols end-to-end: reading digital instructions, applying proper tools, and adapting on the fly, that lie beyond the scope of existing lab automation work.

## 3 AUTOBIO SIMULATOR

### 3.1 AUTOBIO ASSETS

With reference to a real-world biomedical laboratory, AutoBio constructs dimensionally accurate digital models of common biological experimental assets and integrates them into a virtual environment. To support robotic manipulation, AutoBio meticulously preserves the physical interaction properties of these digital assets—including collision characteristics and articulated relationships—to ensure physical fidelity in simulation. All digital assets, along with their associated physical and visual properties, are formally described using the MJCF modeling language.

Table 2: Categories of AutoBio assets

| Type | Function | Asset |
| --- | --- | --- |
| Instrument | Perform experimental procedures | Centrifuge, Thermal Cycler, Mixer, Pipette, . . . |
| Container | Handle and store samples | Centrifuge tube, Cryovial, Pipette tip, . . . |
| Rack | Host containers and tools | Multiple-slot rack, Pipette rack, Tip box |
| Robot | Execute manipulation | UR5e, Aloha, Robotiq gripper, DexHand |

Our assets are broadly categorized into four classes, as listed in Table 2. To ensure the visual and geometrical fidelity of the instrument models, we propose a workflow that incorporates 3D Gaussian Splatting (3DGS) (Kerbl et al., 2023) for modeling laboratory assets. As illustrated in Figure 2, the process begins with multi-view video capture of real-world instruments. We then apply the PGSR algorithm (Chen et al., 2024) to reconstruct high-quality 3DGS assets. Coarse 3D meshes are subsequently extracted to achieve surface reconstruction. The raw mesh from 3DGS often contains redundant vertices and irregular topology due to the discrete nature of Gaussian splats. To optimize the mesh for physical simulation, we refine it in CAD modeling software, producing a smoothed,

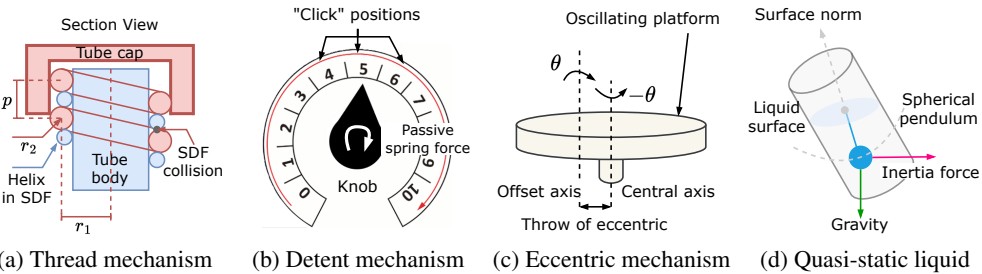

(a) Thread mechanism   (b) Detent mechanism   (c) Eccentric mechanism   (d) Quasi-static liquid

Figure 3: AutoBio physics plugins

watertight low-poly version while preserving critical geometric features with annotated articulations. The refined model is exported in the glTF format, and then is processed with our custom automated tools developed to accelerate the pipeline. First, the **texture generation tool** UV-unwraps the mesh and bakes vertex colors from the high-poly source mesh onto a UV texture map, with features such as automatic seam-aware padding, lighting normalization, and masking for unmatched regions. Second, the **gltf2mjcf converter** transforms the textured and joint-annotated glTF model into a simulation-ready MJCF file for MuJoCo via a lightweight configuration interface. More information on AutoBio assets are presented in the Appendix, including the set of simulation-ready assets we constructed and the implementation details of the automated tools.

## 3.2 AUTOBIO PHYSICS

AutoBio employs MuJoCo as its physics engine to simulate rigid-body dynamics, including interactions among robots, labware and instruments. This framework supports accurate multi-articulated system modeling with rich contact, which are essential for replicating laboratory operations such as pipetting, tube handling, and instrument manipulation. In addition to MuJoCo's native physics features, AutoBio incorporates customized plugins to efficiently simulate laboratory-specific physical behaviors. As depicted in Figure 3, these plugins include:

**Thread mechanism** models the assembly and mechanical properties of mating threads (e.g., the cap-tube assemblies of centrifuge tubes). Inspired by the separation design of collision and visual geometry in MuJoCo, we propose to use *circular helix's* signed distance function (SDF) to substitute thread meshes in the collision detection, a different and more efficient approach compared to prior works (Narang et al., 2022; Jiang et al., 2024). An approximate method for computing this helix SDF is provided in the Appendix. As illustrated in Figure 3a, the collision between the tube and the cap is induced by a pair of coaxial helical threads with identical pitch. Leveraging MuJoCo's SDF-based collision solver, we are able to simulate screw motion between threaded objects as well as the phenomenon of frictional self-locking (through appropriate friction coefficient settings). In contrast to mesh-based collision detection, the SDF approach is agnostic to the convexity of shapes and therefore avoids the need for convex decomposition of thread geometries. This substantially reduces the computational burden of collision handling while preserving high physical fidelity.

**Detent mechanism** simulates incremental motion with discrete "click" positions in lab instruments, such as stepped knobs or handles. Specifically, the detent mechanism provides tactile feedback via passive spring force generated through relative displacement with the nearest gear position.

**Eccentric mechanism** generates oscillating motion through off-axis rotation, enabling realistic simulation of mixers (e.g., vortex mixer). The plugin achieves eccentric orbital motion through negatively coupled rotations about two parallel joint axes.

**Quasi-static liquid** provides an approximate yet efficient simulation of liquid shape deformation within containers, complementing MuJoCo's limitation in fluid modeling. Specifically, this module treats the liquid surface as a planar interface, neglecting wave propagation and pouring effects. This simplification allows us to describe liquid deformation using only two states: the liquid level height and the surface normal vector. The motion of the surface normal is governed by a damped spherical pendulum system in reponse to external container acceleration. We derive the ODE system

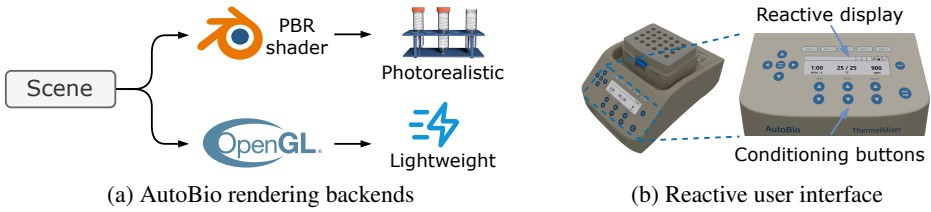

(a) AutoBio rendering backends       (b) Reactive user interface

Figure 4: AutoBio rendering features

of the surface normal through analytical mechanics and Euler-Lagrange equation (see appendix for details). After the normal vector acquired, the liquid body is computed as the intersection of the oriented halfspace and container inner surface. We compute the surface height through volume conservation constraint, thereby generating liquid geometry within the container.

### 3.3 AUTOBIO RENDERING

AutoBio adopts a flexible rendering strategy, offering two rendering backends to accommodate both fast visualization and photorealistic rendering, as summarized by Figure 4a. Our simulation features are equally supported by two backends, but with different visual fidelity. Additionally, it implements real-time texture rendering for reactive panel interfaces in laboratory instruments.

**Basic render** directly utilizes MuJoCo's native OpenGL renderer, which offers fast but limited visualization. Rendering artifacts would occur when visualizing nested transparent objects, such as transparent tubes containing liquids, due to depth sorting errors. So, we fallback to mark some surfaces as opaque for more meaningful results.

**Advanced render** bridges MuJoCo simulation state to Blender's rendering pipeline, leveraging its physically based rendering (PBR) shaders to achieve photorealistic materials grounded in real-world light behavior. Particularly for container-associated manipulation tasks, this solution enables accurate rendering of transparent materials—including polyethylene, glass, and liquids—through configurable optical parameters such as transmission coefficients, refractive indices, and surface roughness.

**Reactive user interface** employs dynamically loaded texture maps to render control panels and displays for some lab instruments, providing visual feedback for robotic manipulation, as shown in Figure 4b. This module maintains full compatibility with both rendering backends described previously.

### 4 AUTOBIO BENCHMARK

Building upon the capabilities of the AutoBio simulator, we develop the AutoBio benchmark, a suite of biologically grounded tasks designed to evaluate robotic automation in laboratory settings.

### 4.1 TASK GENERATION

AutoBio benchmark tasks are defined through a unified procedure that specifies 1). randomized scene initialization; 2). procedural demonstration generation; and 3) task evaluation, with each step expanded below.

**Randomized Scene Initialization.** For each task, the environment is initialized with randomized parameters to enhance diversity of generation. Randomization typically includes variations in the robot's initial joint angles and in the spatial placement of task-relevant objects (e.g., different initial and target positions of centrifuge tubes in the transfer task). Additional domain randomization is supported both in physics (e.g., injected control noise) and visualization (e.g., color and lighting), with detailed task-specific settings provided in Appendix B.2.

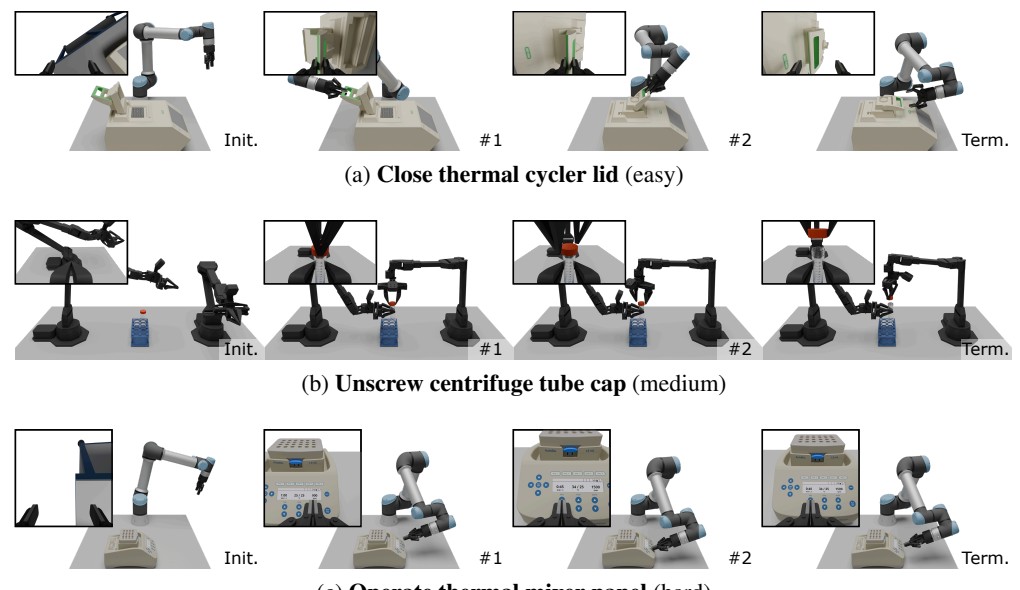

(a) **Close thermal cycler lid** (easy)

(b) **Unscrew centrifuge tube cap** (medium)

(c) **Operate thermal mixer panel** (hard)

Figure 5: Task progression across three difficulty levels. Each step includes a bordered inset (top-left) showing supplementary camera perspectives for contextual clarity.

**Procedural Demonstration Generation.** Expert demonstrations are generated via procedural policies that decompose each task into sequential subtasks (e.g., reach–grasp–lift in the "Pick up centrifuge tube" task). For each subtask, we define an end-effector motion path conditioned on the subtask's initial state and annotated object keypoints. The corresponding joint-space trajectories are computed using inverse kinematics combined with time-optimal path parameterization (TOPP), and the resulting motions are executed via PD control. Subtasks are then concatenated to form a complete expert trajectory.

**Task Evaluation.** Task progress and success are determined through predefined status checks. These include monitoring contact events, object poses, and task-specific metrics. Trajectories that fail to meet these success criteria are discarded and not recorded for training.

## 4.2 TASK CATEGORY

To systematically evaluate VLA models, AutoBio consists of 16 tasks that are categorized into three difficulty levels (Figure 5), characterized by progressively increasing demands on precision, language understanding, and visual reasoning:

- **Level 1 (Easy):** These tasks feature low demands on vision and manipulation precision, exemplified by straightforward actions like closing a thermal cycler lid (5a). Task requirements and language instructions are generally static, making them suitable for initial system integration and basic policy verification.
- **Level 2 (Medium):** Tasks at this level, such as unscrewing a centrifuge tube cap (5b), present increased challenges in visual perception and manipulation precision. Language instructions may vary based on randomized task parameters (e.g., target position), requiring models to exhibit basic generalization and instruction following capability beyond memorization. These tasks are designed to evaluate fundamental VLA model performance.
- **Level 3 (Hard):** High-demand tasks like operating a thermal mixer panel (5c) require visual reasoning, fine-grained manipulation, and robust instruction interpretation. Successful completion often necessitates effective closed-loop control and the ability to perform complex cross-modal reasoning. These tasks are intended to rigorously challenge the capabilities of state-of-the-art VLA models in scientifically relevant scenarios.

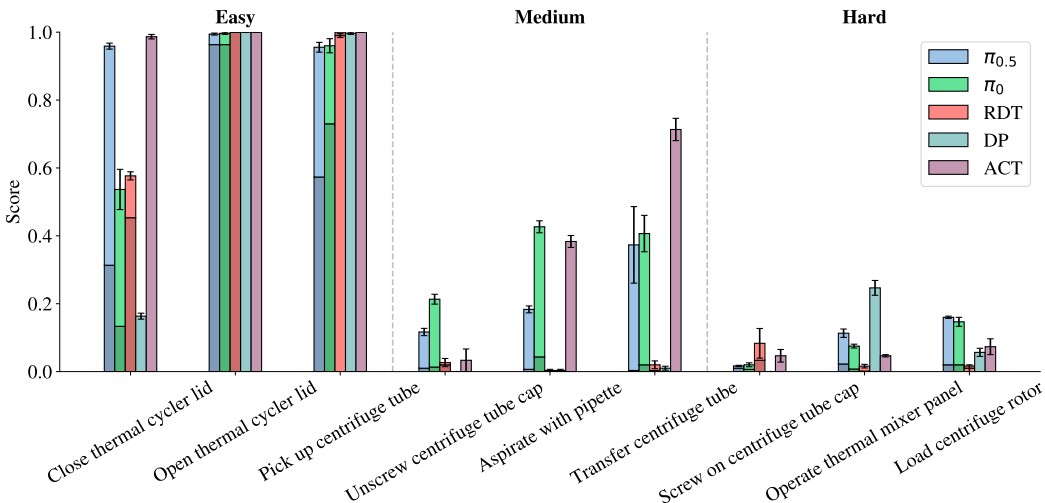

Figure 6: Evaluation scores of VLA and baseline imitation-learning methods on AutoBio tasks of increasing difficulty, trained on 100 demonstrations. Each bar represents the mean score over 100 evaluation episodes, averaged across three runs, with standard error of the mean shown as error bars. For VLA methods, a dimmed stacked segment indicates the smaller-scale (trained on 20 demonstrations) data variant. The full numerical results are provided in Appendix Table 4 and 5.

## 5 EXPERIMENTS

### 5.1 BASELINES AND EXPERIMENTAL SETTINGS

We choose 3 tasks from each difficulty level (Table 4) to evaluate VLA capabilities. For each task, we generate 100 demonstration trajectories at a frequency of $50\,\mathrm{Hz}$ formatted as a LeRobot (Cadene et al., 2024) dataset. The total training set consists of over 792k frame of data, equivalent to 4.4 hours of continuous recording. All tasks except **Operate thermal mixer panel** are scored binarily (1 for success, 0 for failure). For **Operate thermal mixer panel**, we use a relative progress score to better reflect performance difference due to observed policy difficulties. Task details are provided in the appendix.

We evaluate three open-source VLAs: $\pi_{0.5}$ (Black et al., 2025), $\pi_0$ (Black et al., 2024) and RDT (Liu et al., 2025). We adapt our data to each model's requirements (proprioception dimensions, image history, normalization), then finetune with default fine-tuning configurations starting with their pretrained checkpoints ($\pi_{0.5}$-base, $\pi_0$-base and RDT-1B). Adaptation details and model differences can be found in the appendix.

To examine data scaling effects, we train each model on both full (100 episodes) and reduced (20 episodes) datasets while maintaining consistent normalization. Each configuration undergoes three seeded runs, trained for 30000 steps with batch size 32. A single run takes between 10 to 14 hours on an NVIDIA H800 GPU, resulting in a total runtime of approximately 2000 GPU hours.

In addition to the main VLA experiments, we conduct two supplementary experiments about imitation learning baselines and long-horizon task setting, with experimental settings and results detailed in subsection 5.3 and appendix B.5 respectively.

### 5.2 EXPERIMENTAL RESULTS

Policies are evaluated over 100 episodes per task, with aggregated scores presented in Figure 6. The structured visualization highlights several consistent trends across difficulty levels and task families.

**Model-level comparison.** Neither VLA demonstrates universal superiority, but distinct patterns emerge. RDT shows more stable performance on *easy* manipulation tasks, whereas $\pi_0$ excels on *medium* and *hard* tasks requiring visio-linguistic grounding or multi-step precision (e.g., **Operate thermal mixer panel**). The more recent $\pi_{0.5}$ performs similarly to $\pi_0$ overall; its primary improvement is a substantial boost on **Pick up centrifuge tube**, pushing all easy tasks near 100% success. Notably, $\pi_{0.5}$ does *not* close the performance gap on medium/hard tasks, reinforcing that the failure modes we identify persist across generations.

These differences likely stem from architectural design: the PaliGemma-style joint-attention backbone and fully trainable parameters of the $\pi$-series allow richer cross-modal adaptation than RDT's frozen pretrained encoders, which limits its flexibility in visually dense or instruction-heavy tasks.

**Effect of data scaling.** The $\pi$-series models benefit noticeably from additional demonstrations across most tasks, consistent with their $\sim$3B trainable-parameter capacity. In contrast, RDT displays minimal improvement with more data, reflecting the limited adaptability of its static encoders.

**Task difficulty patterns.** The empirical difficulty ordering aligns with our benchmark design. Easy-task failures primarily involve minor gripper slippage or end-effector drift. Medium and hard tasks expose deeper limitations in precision and consistency: tasks involving screwing/unscrewing motions accumulate compounding errors, suggesting that open-loop imitation is insufficient and may require reinforcement learning or feedback-conditioned policies.

**Failure modes and their causes.** Qualitative inspection reveals interpretable failure modes: (1) **Cross-modal grounding errors** occur in **Transfer centrifuge tube**, where models choose incorrect slots, and in **Operate thermal mixer panel**, where textual instructions must be grounded to small UI elements. Low input resolution imposed by current VLA preprocessing pipelines further obscures numeric readouts, leading to inconsistent target selection and oscillatory learning curves. (2) **Visual-reasoning limitations** arise in **Aspirate with pipette** (liquid-level reasoning) and **Load centrifuge rotor** (symmetry maintenance). Memoryless architectures struggle when critical visual cues leave the camera view, motivating future work on temporally coherent or chain-of-thought visual reasoning. (3) **Lack of closed-loop recovery** is evident in contact-rich tasks (e.g., **Screw/Unscrew cap**), where policies often commit to open-loop trajectories and fail to adjust upon minor deviations, indicating the need for adaptive or feedback-driven control.

Together, these quantitative and qualitative findings confirm that AutoBio exposes precision, grounding, and multi-step reasoning gaps that are not captured by existing domestic-task benchmarks.

## 5.3 IMITATION LEARNING BASELINES

To contextualize VLA performance, we evaluate two smaller imitation-learning baselines: Diffusion Policy (DP, $\sim$262M parameters Chi et al. (2023)) and Action Chunking Transformer (ACT, $\sim$52M parameters Zhao et al. (2023)). Since DP and ACT lack language inputs, task-specific parameters extracted from the textual prompts (e.g., target slot indices or mixer settings) are provided as structured numeric observations.

Figure 6 and Appendix Table 5 summarize their results. Both DP and ACT reach competitive performance on several low- and medium-difficulty tasks, demonstrating that high-level language grounding is not required for all settings. However, these baselines struggle to scale to the *full diversity* of AutoBio tasks. The primary obstacle is the heterogeneity of state spaces across tasks (e.g., $[\text{row}, \text{col}]$ for **Transfer centrifuge tube** versus $[\text{rpm}, \text{temp}, \text{time}]$ for **Operate thermal mixer panel**), which prevents training a unified model.

In contrast, VLA models use language as a shared representation of task goals and environment states, enabling a single policy to generalize across radically different task structures. This unified interface suggests stronger potential for multi-task scalability and, ultimately, for executing multi-step laboratory protocols in long-horizon settings.

## 6 Summary

This paper introduce AutoBio, a simulation framework and benchmark designed to evaluate robotic automation in biology laboratory environments. By extending laboratory asset modeling, physics simulation, and rendering capabilities, the AutoBio simulator streamlines the biological experiment simulation inspired by practical operation primitives. Built upon this simulator, we propose the AutoBio benchmark with manipulation tasks across three levels of difficulty, to systematically assess the capabilities of vision-language-action (VLA) models. The experimental results show critical limitations of current VLA models in precision manipulation, instruction following and visual reasoning, and suggest potential improvements in model architecture and training methodologies.

### Acknowledgement

This paper is partially supported by the National Key R&D Program of China No.2022ZD0161000 and the General Research Fund of Hong Kong No.17208825.

### Ethics Statement

This research adheres to the ICLR 2026 ethical guidelines and upholds the principles of responsible research. We ensure that no personally identifiable, sensitive, or harmful data were used. Our experiments did not involve any human subjects vulnerable groups. We have considered the potential societal impact of our methods, including the risk of misuse, and believe that these contributions primarily advance scientific understanding and do not pose foreseeable harm.

### Reproducibility Statement

We follow the reproducibility guidelines in the ICLR 2026 author guidelines. We will open source code, configuration files, and scripts to reproduce our results, including dataset construction, model training, and evaluation, on platforms such as GitHub and Huggingface as soon as possible.

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

## A AUTOBIO SIMULATION

### A.1 AUTOBIO ASSETS OVERVIEW

Figure 7 presents the collection of simulation-ready digital assets included in AutoBio.

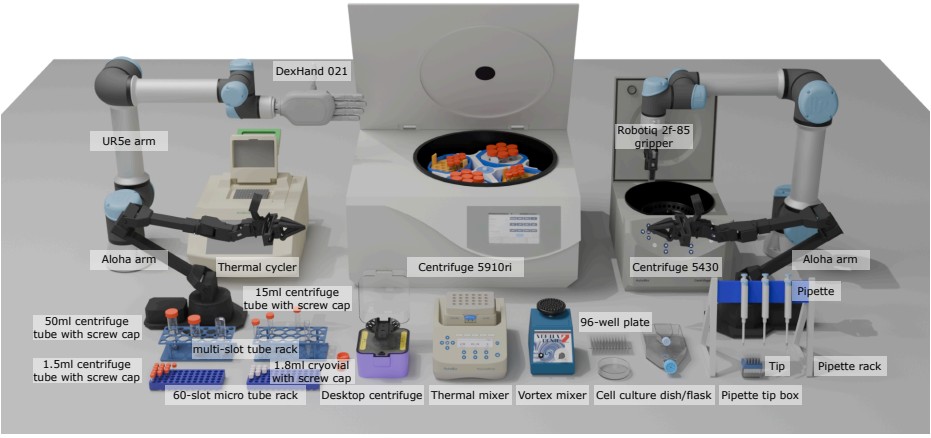

Figure 7: The overview of AutoBio's simulation-ready digital assets.

## A.2 IMPLEMENTATION DETAILS OF ASSETS DEVELOPING TOOLS

**Texture generation tool**. The meshes extracted from 3DGS describe color information with vertex color, which cannot be applied directly to the refined meshes. This tool handles this issue by transferring vertex colors to UV texture maps by ray casting, with features including seam-aware padding, lighting normalization (color correction), and masking of unmatched regions. This produces a texture with reasonable out-of-box appearance, as well as allowing subsequent manual editing.

**gltf2mjcf converter** is a utility we developed to streamline the integration of CAD-designed assets into MuJoCo. It automatically transforms joint-annotated glTF models—exported from SolidWorks or Blender—into simulation-ready MJCF files through a lightweight configuration interface. This eliminates the need for manual MJCF editing and significantly accelerates the process of preparing complex articulated models for physics simulation.

A configuration file specifies global model metadata (e.g., name, scale, timestep options) as well as body hierarchies, joint definitions, and actuator properties. Each section corresponds to a glTF node and may optionally include additional parameters such as damping, stiffness, or control ranges, thereby providing a flexible mechanism for defining both kinematics and dynamics. A sample file in `toml` style is provided below:

```
[__meta__]
name = "vortex_mixer_genie_2"
scale = 76.25   # 0.0016 m -> 0.122 m
option = {timestep="0.001"}

[body]
# Vortex mixer body
[platform]
# Rubber platform: 600-3200 RPM -> 62.83-335.10 rad/s
parent = "body"
[platform.joint]
type = "orbital"
reference = "platform-joint"
offset = [0, -0.002, 0]  # in body frame
extra = {damping="0.01", armature="0.1"}
actuator = { kind="velocity", kv="1", ctrlrange="0 400" }
[switch]
# Switch: on/off/touch (truncated)
[knob]
# Knob: speed control 0~10 (truncated)
```

**Asset creation time analysis**. We report the typical time required to build a single simulation-ready asset with our proposed workflow. The process begins with approximately 3 minutes of manual video capture, followed by about 40 minutes of automated 3DGS training and mesh extraction. Subsequent CAD refinement and joint annotation, performed by experienced users, generally take 30 to 60 minutes depending on model complexity. The remaining downstream automated steps, including texture generation and MJCF conversion, require around 3 minutes.

## A.3 THREAD MECHANISM

A parameterized circular helix in three-dimensional space can be mathematically expressed as:

$$H(t; r_1, p) = [x, y, z]^\top = [r_1 \cos t, r_1 \sin t, pt]^\top.$$

In practical applications, a helix is always bounded. For the above helix formulation, we denote these bounds $2\pi l \leq t \leq 2\pi h$, where $l$ and $h$ represent the starting and ending thread counts measured

from the zero position. Within this framework, the approximate SDF becomes:

$$t_0 = \text{atan2}(P_y, P_x),$$

$$k = \left\lfloor \frac{P_z - t_0 p}{2\pi p} \right\rceil, \quad l' = \left\lceil \frac{l - t_0}{2\pi} \right\rceil, \quad h' = \left\lfloor \frac{h - t_0}{2\pi} \right\rfloor,$$

$$SDF(P) = \begin{cases} d_P(2k\pi + t_0) & l' \le k \le h' \\ \min\{d_P(2\pi l), d_P(2l'\pi + t_0)\} & k < l' \\ \min\{d_P(2\pi h), d_P(2h'\pi + t_0)\} & h' < k \end{cases},$$

where $d_P(t)$ represents the Euclidean distance between point $P$ and the corresponding position on the helix parameterized by $t$:

$$d_P(t) = \|H(t; r_1, p) - P\|_2.$$

This approximation remains valid when the helix angle is sufficiently small.

In practical implementation, the plugin has been adapted to MuJoCo as an `sdf` extension. While alternative methods exist to approximate screw motion—such as parenting the nut to the bolt within a kinematic tree and adding a coupled hinge joint and slide joint constrained by screw motion—these approaches present two significant limitations: (1) The self-locking mechanism does not naturally emerge from such couplings and proves challenging to implement artificially; (2) Modeling transitional states between fully coupled and completely free bolt-nut pairs becomes problematic. By employing SDF to characterize these interactions in a manner more faithful to real-world physics, we address both challenges gracefully.

## A.4 DETENT MECHANISM

The detent mechanism is modeled using a passive force function $f(q, \dot{q})$ that depends exclusively on the generalized position and velocity—states intrinsic to any dynamic system. For a knob with $n$ discrete positions, the force calculation follows:

$$f(q, \dot{q}) = -k(q - q_j) - \lambda \dot{q},$$

where the target position index is determined by:

$$j = \underset{i}{\text{argmin}} |q - q_i|, \quad i = 0, \ldots, n-1.$$

In practical implementation, this mechanism is adapted to MuJoCo as a `passive force` extension.

## A.5 ECCENTRIC MCHANISM

Planar eccentric motion can be mathematically represented through a planar transformation matrix, which admits the following decomposition:

$$\begin{pmatrix} 1 & 0 & t\cos\theta \\ 0 & 1 & t\sin\theta \\ 0 & 0 & 1 \end{pmatrix} = \begin{pmatrix} \cos\theta & -\sin\theta & t\cos\theta \\ \sin\theta & \cos\theta & t\sin\theta \\ 0 & 0 & 1 \end{pmatrix} \begin{pmatrix} \cos\theta & \sin\theta & 0 \\ -\sin\theta & \cos\theta & 0 \\ 0 & 0 & 1 \end{pmatrix}, \quad (1)$$

where $t$ denotes the throw of the eccentric. The right-hand side matrices correspond to the motions of two negatively-coupled hinge joints. Instruments employing eccentric mechanisms, such as vortex mixers, typically operate at high RPMs. Under these conditions, coupling two joints through equality constraints proves more robust than alternative solutions.

## A.6 QUASI-STATIC LIQUID

The surface normal dynamics are governed by a damped spherical pendulum system. Following analytical mechanics conventions, we present the system's Lagrangian and generalized force as:

$$L = \frac{1}{2}ml^2\left(\dot{\phi}^2 \sin^2\theta + \dot{\theta}^2\right) + ml\left(g_x \sin\theta\cos\phi + g_y \sin\phi\sin\theta - g_z\cos\theta\right),$$

$$Q = [-\lambda_\phi\dot{\phi}, -\lambda_\theta\dot{\theta}]^\top.$$

$[\phi, \theta]$ are the spherical coordinates of the normal vector and the generalized coordinates of the system. $l$ represents the characteristic length, and $m$ is system mass (which ultimately cancels out in the final equations). The vector $g = [g_x, g_y, g_z]^\top$ captures time-varying accelerations resulting from both gravity and inertial forces. $\lambda_\phi, \lambda_\theta$ are configurable damping coefficients. Applying the Euler-Lagrange equation:

$$\frac{\mathrm{d}}{\mathrm{d}t}\frac{\mathrm{d}L}{\mathrm{d}\dot{q_i}} - \frac{\mathrm{d}L}{\mathrm{d}q_i} = Q_i,$$

we derive the following system of ordinary differential equations:

$$\frac{\mathrm{d}}{\mathrm{d}t}\phi = \dot{\phi},$$

$$\frac{\mathrm{d}}{\mathrm{d}t}\theta = \dot{\theta},$$

$$\frac{\mathrm{d}}{\mathrm{d}t}\dot{\phi} = \frac{-2ml^2 v_\phi v_\theta \sin\theta \cos\theta + ml\left(-g_x \sin\phi + g_y \cos\phi\right)\sin\theta - \lambda_\phi v_\phi}{ml^2 \sin^2\theta},$$

$$\frac{\mathrm{d}}{\mathrm{d}t}\dot{\theta} = \frac{ml^2 v_\phi^2 \sin\theta \cos\theta + ml\left(g_x \cos\phi \cos\theta + g_y \sin\phi \cos\theta + g_z \sin\theta\right) - \lambda_\theta v_\theta}{ml^2}.$$

In practical implementation, to avoid numerical instabilities when the system approaches simple pendulum behavior, we modify the denominator in the $\frac{\mathrm{d}}{\mathrm{d}t}\dot{\phi}$ equation to $ml^2 \max\{\sin^2\theta, \epsilon\}$. The system initializes with states aligned to gravity direction and zero velocity. The normal direction components are computed as:

$$x = \sin\theta \cos\phi,$$
$$y = \sin\theta \sin\phi,$$
$$z = -\cos\phi.$$

Following surface normal determination at each timestep, we compute the liquid body as the intersection between the oriented halfspace and the container's inner surface. This implementation involves computing triangle-plane relations for the manifold mesh. Surface height computation proceeds through volume conservation constraints, employing the previous height as an initial guess before refining via Newton-Bisect search.

### A.7 RENDERING

The MuJoCo renderer utilizes the legacy OpenGL fixed-function pipeline for scene rendering, which offers limited customization capabilities. Notably, its reliance on fixed-function lighting creates visual inconsistencies between textured and untextured surfaces under specular and ambient lighting conditions. For this renderer, we provide best-effort support, with certain scene elements (such as liquid representations) appearing in simplified forms to minimize visual artifacts.

We develop a bridging mechanism to translate MuJoCo's scene definition (`MjModel`) and simulation state (`MjData`) into Blender's environment. While conceptually similar to MuJoCo's ongoing USD (Universal Scene Description) exporter, our solution specifically accommodates our workflow by enabling: (1) Importation of tuned assets from Blender gallery files, and (2) Direct generation of fully configured Blender scenes capable of rendering features beyond MuJoCo's representational capacity.

### A.8 REACTIVE UI

Our user interfaces currently operate in retained mode, meaning it only repaints when changes occur. For the MuJoCo renderer, UI updates in the 3D environment are achieved through calls to `mjr_uploadTexture` following repainting. In the Blender renderer, the UI is passed to the `Image Texture` node as either static images or video streams.

## B EXPERIMENT DETAILS

### B.1 ROBOT CONFIGURATION

The AutoBio benchmark currently supports two primary robotic arm configurations:

- **Aloha:** This arm is equipped with its native gripper. Characterized by a smaller reach and lower payload capacity, it is primarily employed in tasks involving lighter objects, such as handling individual centrifuge tubes.

- **UR5e:** This arm offers greater reach and payload. We provide two end-effector options for the UR5e: (a) **UR5e-Robotiq:** The UR5e paired with a Robotiq 2F-85 parallel gripper. This configuration is suited for tasks requiring interaction with larger instruments or a wider operational range. (b) **UR5e-DexHand:** The UR5e equipped with a DexHand 021, a 19-DOF dexterous hand. This setup is intended for tasks demanding more dexterity, such as precise pipette operation. Recognizing that current VLA models are often optimized for simpler, low-DOF end-effectors, we offer a simplified control mode for the DexHand. In this mode, most of its DOFs are pre-configured and fixed, with only the metacarpophalangeal (MCP) joint of the thumb actuated collectively to mimic a gripper-like open/close action.

## B.2 BENCHMARK TASKS

In the experiment section, we evaluated VLA capabilities across 9 AutoBio tasks. Below we describe the details of each task, as well as additional tasks not present in the main experiment.

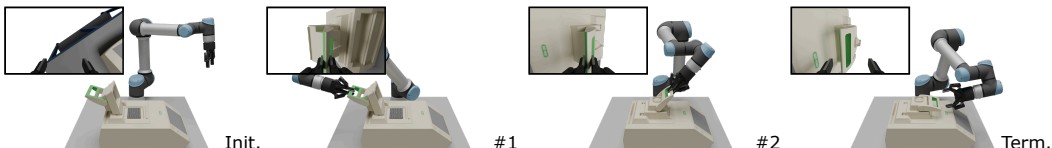

**Close thermal cycler lid** (easy, ∼20 s): *"close the lid of the thermal cycler"* Close and lock the thermal cycler lid using a *UR5e-Robotiq* robot, testing trajectory following for articulated object manipulation. The joint angles of the robotic arms are perturbed to add randomness to the task. This randomization also applies to all tasks below.

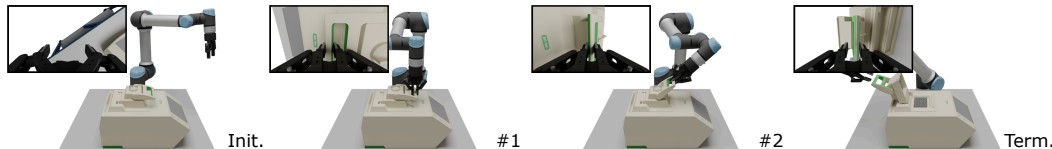

**Open thermal cycler lid** (easy, ∼17 s): *"open the lid of the thermal cycler"* Unlock and open the thermal cycler lid using a *UR5e-Robotiq* robot, testing trajectory following for articulated object manipulation.

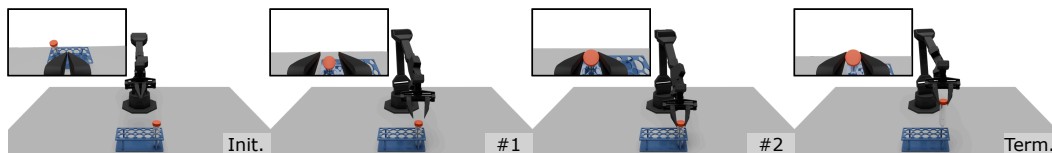

**Pick up centrifuge tube** (easy, ∼10 s): *"pick up the centrifuge tube on the rack"* Pick up a centrifuge tube from its rack using an *Aloha* robot, focusing on basic visual positioning. The tube is placed in a random rack slot to promote generalization. This also applies to all tasks below involving tubes and racks.

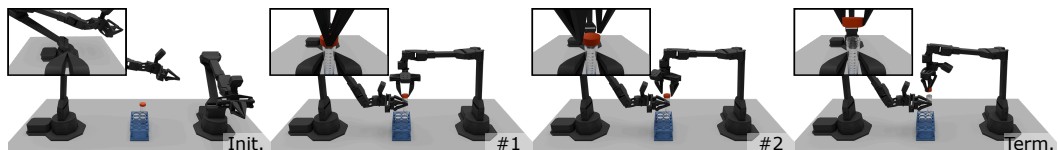

**Unscrew centrifuge tube cap** (medium, ∼27 s): *"dual-Aloha arms unscrewing centrifuge tube cap: one stabilizes tube while the other twists cap"* Remove the centrifuge tube cap using two *Aloha* robots. This evaluates dual-arm coordination and manipulation precision.

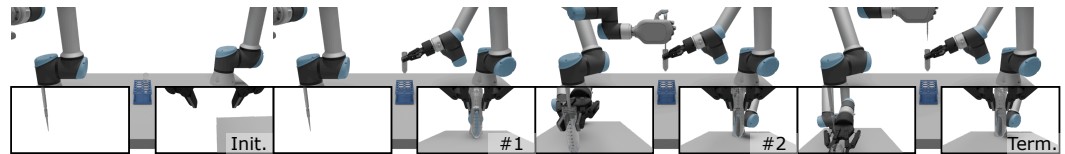

**Aspirate with pipette** (medium, ∼14 s): *"dual-UR5e pipetting: one arm lifts centrifuge tube, the other aligns pipette tip and aspirates liquid"* Aspirate liquid from a tube using a *UR5e-Robotiq* (holder) and *UR5e-DexHand* (pipette operator). This tests dual-arm coordination, precision, and visual reasoning. The liquid volume in the tube is randomized to test liquid level sensing capabilities.

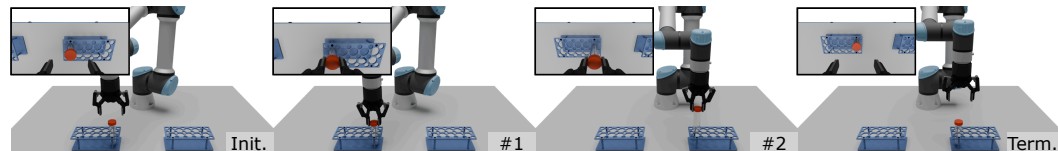

**Transfer centrifuge tube** (medium, ∼11 s): *"pick up the centrifuge tube and move it to the other rack, row {target_row}, column {target_col}"* Transfer a tube to a specified rack slot using a *UR5e-Robotiq* robot, requiring precise manipulation, visual positioning, and language instruction following. The target rack slot is randomized to verify instruction following.

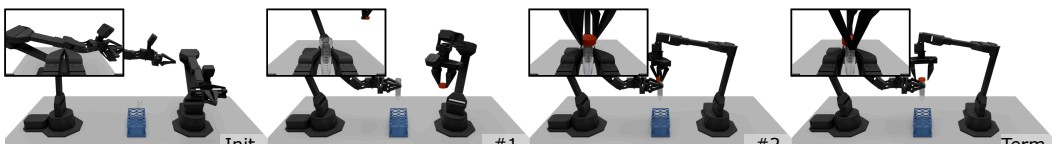

**Screw on centrifuge tube cap** (hard, ∼31 s): *"dual-Aloha arms screwing on centrifuge tube cap: one grips tube while the other twists cap"* The inverse of unscrewing, but with stricter precision requirements for proper alignment.

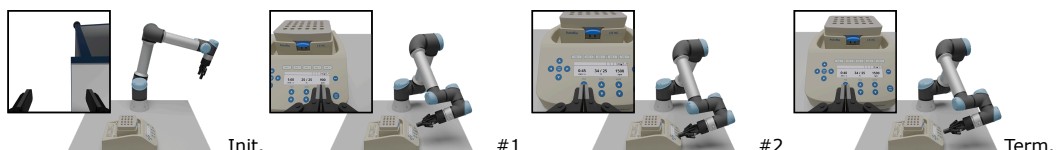

**Operate thermal mixer panel** (hard, ∼16 s): *"Adjust thermal mixer parameters, with speed set to {set_rpm} rpm, temperature set to {set_temp} °C, and time set to {set_time} seconds"* Set parameters (time, temperature, frequency) on a mixer panel using a *UR5e-Robotiq* robot following language instruction and UI feedback, evaluating visual reasoning, language understanding, and high-precision manipulation. The parameters are randomized to test cross-modal reasoning.

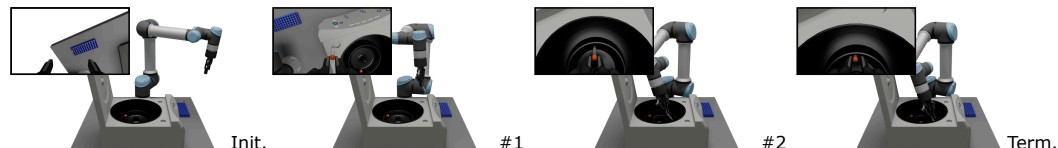

**Load centrifuge rotor** (hard, ∼11 s): *"Insert a second centrifuge tube into the slot that is symmetrically opposite to the currently placed tube"* Load a tube into the correct rotor slot while maintaining symmetry with the existing tube, testing advanced visual reasoning and precise positioning. The target rotor slot are randomized by setting the currently placed tube to different slots, and the rotor angle are also randomized.

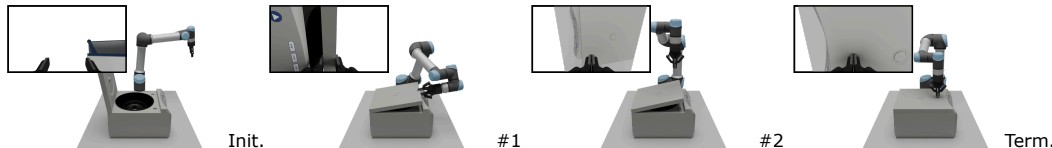

**Close centrifuge lid** (easy, ∼15 s): *"Close the lid of the centrifuge"* Close and lock the centrifuge lid using a *UR5e-Robotiq* robot, testing multi-step trajectory following for articulated object manipulation.

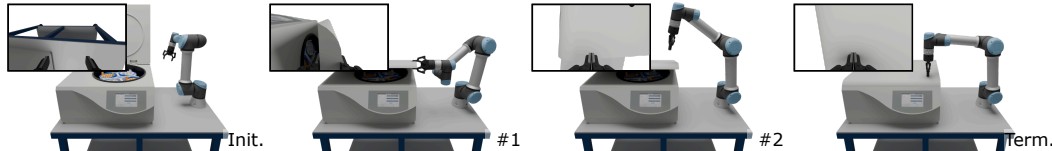

**Close large centrifuge lid** (easy, ∼15 s): *"Close the lid of the centrifuge"* Close and lock the centrifuge lid using a *UR5e-Robotiq* robot, testing multi-step trajectory following for articulated object manipulation.

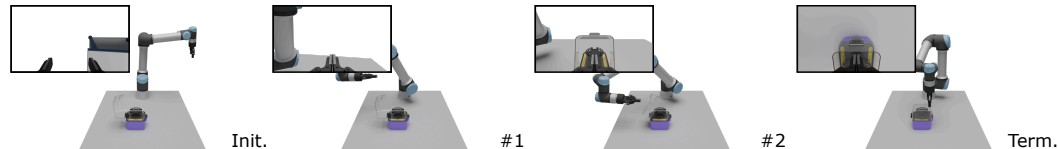

**Close mini centrifuge lid** (easy, ∼10 s): *"Close the lid of the centrifuge"* Close the centrifuge lid using a *UR5e-Robotiq* robot, testing trajectory following for articulated object manipulation.

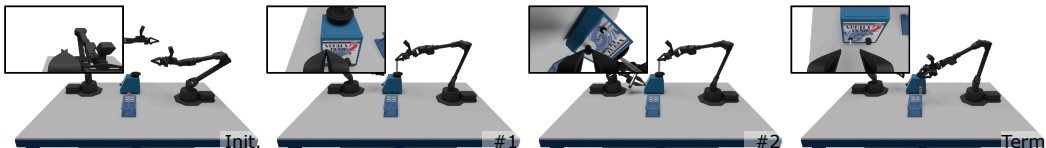

**Vortex mix centrifuge tube** (hard, ∼60 s): *"dual-Aloha arms vortex mix the centrifuge tube on the vortex mixer: one holds the tube, the other operates the vortex mixer to gear {gear}"* Vortex mix a centrifuge tube with two *Aloha* robots. This experiment comprehensively tests dual arm coordination, articulated object handing, UI reading, and long-horizon performance.

The evaluation time limits extended by approximately $50\%$ compared to demonstration horizon to accommodate imperfect policy execution. All tasks except **Operate thermal mixer panel** are scored binarily (1 for success, 0 for exceeding time limit) based on pose and contact requirements at checkpoints and terminal states. For **Operate thermal mixer panel**, we implement a weighted relative progress score to better reflect performance difference due to observed policy difficulties:

$$\text{Score} = \sum_{i=1}^{3} w_i \max\left\{1 - \frac{|\text{final}_i - \text{target}_i|}{|\text{initial}_i - \text{target}_i|}, 0\right\},$$

normalized to $[0, 1]$.

## B.3 BASELINES

We summarize key differences of our baselines ($\pi_0$, $\pi_{0.5}$ and RDT) in Table 3, where some of them (proprioception dimensions, image history, normalization) lead to difference in data adaptation.

Table 3: Key differences between $\pi_0$ and RDT baselines

|  | $\pi_0$, $\pi_{0.5}$ | RDT |
|---|---|---|
| Architecture | Decoder-only | Encoder-decoder |
| Vision backbone | SigLIP So400m/14 224 | SigLIP So400m/14 384 |
| Language backbone | Gemma 2B+300M | T5 v1.1 XXL |
| Trainable weights | Full | V & L backbone frozen |
| Action sampling | Flow matching | Diffusion |
| Normalization | All | Gripper width only |
| State dimension | 32 | 128 |
| Action horizon | 50 | 64 |
| Image history | 0 (None) | 1 |

## B.4 EVALUATION RESULT TABLES FOR THE MAIN BENCHMARK EXPERIMENT

The raw numeric evaluation results for VLA and baseline imitation learning methods are summarized in Table 4 and Table 5 respectively.

Table 4: VLA evaluation score measured over 100 episodes on AutoBio tasks with increasing level of difficulty. The score is reported in percentage by computing the value and the standard error of the mean over three runs.

| # Demo | 20 | 100 | 20 | 100 | 20 | 100 |
|---|---|---|---|---|---|---|
| | | | **Easy level** | | | |
| | **Close thermal cycler lid** | | **Open thermal cycler lid** | | **Pick up centrifuge tube** | |
| $\pi_{0.5}$ | $96.3 \pm 0.9$ | $99.4 \pm 0.3$ | $57.3 \pm 0.9$ | $95.6 \pm 1.4$ | $31.3 \pm 1.2$ | $95.9 \pm 0.9$ |
| $\pi_0$ | $96.3 \pm 0.7$ | $99.7 \pm 0.3$ | $73.0 \pm 3.1$ | $96.0 \pm 2.1$ | $13.3 \pm 0.9$ | $53.7 \pm 5.9$ |
| RDT | $100.0 \pm 0.0$ | $100.0 \pm 0.0$ | $100.0 \pm 0.0$ | $99.0 \pm 0.6$ | $45.3 \pm 6.8$ | $57.7 \pm 1.2$ |
| | | | **Medium level** | | | |
| | **Unscrew centrifuge tube cap** | | **Aspirate with pipette** | | **Transfer centrifuge tube** | |
| $\pi_{0.5}$ | $1.0 \pm 0.6$ | $11.7 \pm 1.1$ | $0.7 \pm 0.3$ | $18.3 \pm 1.0$ | $0.3 \pm 0.3$ | $37.3 \pm 11.3$ |
| $\pi_0$ | $1.3 \pm 1.3$ | $21.3 \pm 1.5$ | $4.3 \pm 0.3$ | $42.7 \pm 1.8$ | $2.0 \pm 0.6$ | $40.7 \pm 5.4$ |
| RDT | $2.0 \pm 1.5$ | $2.7 \pm 1.2$ | $0.0 \pm 0.0$ | $0.3 \pm 0.3$ | $0.3 \pm 0.3$ | $2.0 \pm 1.2$ |
| | | | **Hard level** | | | |
| | **Screw on centrifuge tube cap** | | **Operate thermal mixer panel** | | **Load centrifuge rotor** | |
| $\pi_{0.5}$ | $1.0 \pm 0.0$ | $1.7 \pm 0.2$ | $2.2 \pm 0.4$ | $11.3 \pm 1.2$ | $2.0 \pm 0.6$ | $16.0 \pm 0.3$ |
| $\pi_0$ | $0.7 \pm 0.3$ | $2.0 \pm 0.6$ | $0.8 \pm 0.2$ | $7.5 \pm 0.6$ | $2.0 \pm 0.6$ | $14.7 \pm 1.3$ |
| RDT | $3.3 \pm 1.2$ | $8.3 \pm 4.4$ | $0.2 \pm 0.1$ | $1.6 \pm 0.5$ | $1.7 \pm 1.2$ | $1.0 \pm 1.0$ |

Table 5: Evaluation scores of baseline imitation learning methods measured over 100 episodes on AutoBio tasks with increasing level of difficulty.

| | **Easy level** | | | **Medium level** | | | **Hard level** | | |
|---|---|---|---|---|---|---|---|---|---|
| DP | $100.0_{\pm 0.0}$ | $99.7_{\pm 0.3}$ | $16.3_{\pm 0.9}$ | $0.0_{\pm 0.0}$ | $0.3_{\pm 0.3}$ | $1.0_{\pm 0.6}$ | $0.0_{\pm 0.0}$ | $24.7_{\pm 2.2}$ | $5.7_{\pm 1.2}$ |
| ACT | $100.0_{\pm 0.0}$ | $100.0_{\pm 0.0}$ | $98.7_{\pm 0.7}$ | $3.3_{\pm 3.3}$ | $38.3_{\pm 1.8}$ | $71.3_{\pm 3.3}$ | $4.7_{\pm 1.9}$ | $4.7_{\pm 0.3}$ | $7.3_{\pm 2.3}$ |

### B.5 Additional experiment results for Long-Horizon Task

Human operators naturally decompose complex experiments into atomic subtasks, and execute step by step—a capability we test by combining **Close/Open thermal cycler lid** episodes (200 in total) to evaluate trajectory concatenation. We assess performance by executing one subtask to completion before switching prompts (Table 6).

Table 6: Trajectory concatenation evaluation result

| | **Close** | **Open** | **Close-Open** | **Open-Close** |
|---|---|---|---|---|
| $\pi_{0.5}$ | $99.3 \pm 0.3$ | $95.7 \pm 0.9$ | $3.7 \pm 0.9$ | $15.3 \pm 2.0$ |
| $\pi_0$ | $100.0 \pm 0.0$ | $96.0 \pm 0.6$ | $3.0 \pm 1.2$ | $87.3 \pm 2.7$ |
| RDT | $98.7 \pm 0.3$ | $88.0 \pm 3.2$ | $4.3 \pm 2.8$ | $8.7 \pm 2.2$ |

After training on mixed data, the performance of RDT drops slightly on the two original tasks **Close** and **Open**, while $\pi_0$'s performance mostly remains the same. Regarding concatenated tasks, since the terminal robot pose of **Open** aligns well with **Close**'s trajectory, $\pi_0$ could achieve reasonable transition in **Open-Close**, yet RDT fails to effectively interpolate in-between. **Close-Open** fails more frequently for both models, due to handle grip orientation differences in demonstration at transition location. This suggests current VLAs primarily memorize trajectories during fine-tuning, with limited ability to generalize or smoothly transition between related tasks.

## C   LLM USAGE DISCLOSURE

In accordance with the ICLR 2026 policy on LLM disclosure, we acknowledge the use of LLM in the preparation of this paper. The model was used strictly as a tool to aid and polish the writing. All scientific content is the original work of the authors.

