# OpenReview forum: "AutoBio: A Simulation and Benchmark for Robotic Automation in Digital Biology Laboratory"
_ICLR.cc/2026/Conference — ICLR 2026 Poster_

### Official Review · Reviewer_knDW · 2025-10-29

**Soundness:** 3
**Presentation:** 3
**Contribution:** 3
**Rating:** 6
**Confidence:** 3

**Summary:**

The paper introduces AutoBio, a simulation framework and benchmark aimed at evaluating vision-language-action (VLA) models for biology lab automation. Beyond standard rigid-body interaction, AutoBio adds (i) a digitization pipeline to turn real instruments into simulation-ready assets, (ii) MuJoCo physics plug-ins for lab-specific mechanisms, and (iii) a dual rendering stack (fast MuJoCo + Blender PBR bridge) with reactive instrument UIs. The benchmark contains 16 tasks across three difficulty levels; the experiments train/evaluate π0, π0.5, and RDT on 9 tasks, plus DP/ACT imitation baselines. Results: near-ceiling performance on “easy” tasks; sharp drops on medium/hard tasks involving precision screw motions, UI following, liquid-level reasoning, and rotor symmetry.

**Strengths:**

1. Well-Motivated and Novel Domain: The paper successfully identifies a clear and important gap in current robotics research. Moving from generalist domestic tasks to specialized, high-precision professional domains like a biology lab is a logical and necessary next step for the field. The unique challenges of this domain (precision, transparency, complex tool-use, protocol-following) are well-articulated.
2. New Simulation Features: The authors engineered new capabilities to meet the domain's demands, like an asset digitization pipeline, custom physics plugins.
3. Rigorous Benchmark and Clear Results: The benchmark tasks are well-designed and thoughtfully scaffolded from Easy to Hard. The experimental results are stark and unambiguous: SOTA VLA models fail significantly as task complexity increases. This demonstrates that AutoBio is a challenging and valuable benchmark that is not "solved" and will be effective at driving future research.

**Weaknesses:**

1. Lack of full, long-horizon protocol tasks: The introduction emphasizes "Long-horizon workflows"  as a key challenge. However, the benchmark consists of 16 discrete, relatively short-horizon tasks. The "long-horizon" experiment in the appendix (B.4) is weak, merely concatenating two easy tasks (open/close lid). A true test would be a multi-stage protocol combining several different primitives (e.g., pick tube, unscrew, aspirate, transfer to a new tube, place in centrifuge, operate panel).
2. Evaluation uses 9 of 16 tasks: the main results cover a subset (3 per level). It’s unclear whether conclusions generalize across all 16 tasks; several compelling tasks in Appendix B.2 (e.g., vortex mixing, multiple centrifuge variants) are left unreported. A fuller sweep or cross-task generalization metric would strengthen the case.
3. Analysis of VLA failures: The analysis that imitation learning fails on high-precision tasks due to "compounding errors"  is correct but not particularly novel. The procedural nature of the demonstrations  (i.e., "perfect" kinematic paths) is a known confounding factor for IL, as models struggle to mimic this non-human data. The paper would be stronger if it discussed this limitation or compared performance with teleoperated human data.
4. No sim-to-real validation: the work is framed as a stepping stone to lab automation, yet no real-robot evaluation is presented. Given the Blender PBR bridge, a small real-robot demo (even on easy tasks) would substantially improve credibility.

**Questions:**

1. The "Operate thermal mixer panel" task requires reading a reactive display. The analysis notes that models struggle, in part due to low-resolution inputs obscuring the numbers. Was the "basic" OpenGL renderer or the "advanced" photorealistic Blender renderer used to generate the visual data for training the VLA models?
2. When training a single policy across all 9 (or 16) tasks, does performance degrade vs per-task finetuning?

---

> ### Author Response · Authors · 2025-11-19
> **Response (1/2)**
>
> We sincerely appreciate the reviewer's constructive review and their insightful questions. Below, we address each concern in detail, incorporating additional clarifications and discussions to strengthen the manuscript.
>
> **W1** (long-horizon):
>
> We thank the reviewer for this constructive point. Long-horizon biology protocols are indeed a central motivation for AutoBio. In the current submission, we chose to focus our main experiments on shorter horizon tasks because our demonstration-synthesis infrastructure is not yet mature enough to robustly support flexible, diverse, and error-tolerant long-horizon data generation.
>
> We fully agree that more long-horizon tasks would strengthen the benchmark, and this is a major direction of our ongoing work. We are actively developing: (1) automatic scene composition to facilitate multi-stage protocols, (2) standardized representations for multi-step experimental procedures, and (3) LLM-driven synthesis pipelines that can generate coherent long-horizon demonstrations.
>
> We also note that long-horizon execution presents challenges beyond data generation—for instance, controlled recovery from intermediate failures is unlikely to emerge from large-scale imitation of only successful trajectories. These issues remain open problems in VLA research.
>
> **W2** (task evalution):
>
> Thank you for raising this point. Our choice to report results on 9 of the 16 tasks was primarily to keep the main benchmark evaluation focused and interpretable while avoiding redundancy.
>
> Multiple centrifuge-lid variants test essentially the same capability: short-horizon trajectory following of a rigid-body opening/closing mechanism. In preliminary experiments, these tasks produced failure modes and performance trends that were nearly identical to thermal cycler handling. We therefore included the thermal cycler case in the main text because its locking mechanism is slightly more complex and thus more revealing. We agree that cross-task generalization among such similar variants is interesting; we will incorporate this analysis in future work.
>
> The vortex-mixer task has a much longer (~60 s) and multi-stage horizon. On all methods we tested, success rate was effectively zero even with 100 demonstrations, and errors were dominated by long-horizon drift rather than semantically interesting failures. Because it did not provide meaningful differentiation between methods, we excluded it from the main benchmark until better long-horizon policies are available.
>
> **W3** (compounding error):
>
> We appreciate this observation. We agree that attributing IL failures to "compounding errors" is not new, and that the use of procedurally generated, noise-free trajectories can further amplify this issue. We currently mitigate this through domain randomization (as detailed in the task appendix), where we introduce variation in object placement, robot initialization, and other environmental factors. However, the demonstration trajectories themselves remain relatively "perfect" because our present synthesis pipeline lacks adaptive feedback or force-based correction; injecting large trajectory noise would therefore cause many demonstrations to fail during execution. We acknowledge this as a limitation of the current benchmark and will make it explicit in the revised analysis.
>
> Regarding teleoperation: we agree that human-operated demonstrations often provide richer, more natural variation and could reduce the gap introduced by procedural paths. However, in our domain this presents practical challenges. Many tasks—such as screwing motions or contact-rich instrument operations—require fine-grained force cues and dexterity that are difficult to reproduce with typical master–slave teleoperation setups, especially those without haptic feedback. As a result, there is a trade-off between realism and feasibility.
>
> **W4** (real-world experiment):
>
> We thank the reviewer for highlighting the importance of sim2real transfer evaluation in assessing simulation benchmarks. We fully agree that demonstrating alignment between simulation and real-world performance is an important long-term goal for simulation environments, particularly those intended to support policy learning. However, conducting real-world robotic experiments in a biology lab setting presents unique challenges. Many of the instruments involved are delicate, costly, and require careful handling. Given our current infrastructure, we chose to prioritize simulator fidelity and safety for early-stage development. We plan to revisit sim2real validation as our hardware and automation infrastructure mature.

---

> ### Author Response · Authors · 2025-11-19
> **Response (2/2)**
>
> **Q1** (UI resolution):
>
> The display, technically being a 2D canvas, is rendered with dedicated 2D rendering libraries (Matplotlib in the current implementation, with planned migration to Skia for performance) into a 2D texture. The texture is then uploaded to the chosen 3D rendering backend (OpenGL/Blender) as albedo texture of the display. Therefore, the visual fidelity of the display is not fundamentally different across backends, only the different material-lighting interaction leading to different level of photorealism. And to answer your question concretely, the reported data use OpenGL renderer.
>
> **Q2** (multi-task):
>
> Our paper includes one configuration that is partially multi-task: in Appendix "Additional Experiment Results for Long-Horizon Task", we train a VLA model on a mixture of demonstrations from two subtasks. This experiment shows that the VLA achieves reasonable per-task performance when trained jointly on those tasks. So, as an answer to your question, the policy performance seems not to degrade under such easy multi-task settings.
>
> Regarding broader multi-task evaluations: our demonstration-generation and training pipelines do not impose restrictions on multi-task training. However, current open-source VLAs are typically fine-tuned for a single robotic embodiment—their action heads are neither embodiment-agnostic nor explicitly embodiment-conditioned—so jointly training tasks collected from different robots would be technically inconsistent. For this reason, we did not mix tasks across embodiments in the main paper.
>
> That said, we agree that a stronger multi-task baseline is valuable. In the upcoming revision, we will add an experiment that jointly trains on all uni-manual UR5e tasks to provide a more complete assessment of multi-task performance in a consistent embodiment setting.

---

### Official Review · Reviewer_yC5L · 2025-11-01

**Soundness:** 3
**Presentation:** 3
**Contribution:** 4
**Rating:** 6
**Confidence:** 4

**Summary:**

This paper presents AutoBio, a robotics benchmark in simulation evaluating robotic agents' ability to perform lab tasks/experiments important for biology research. It is motivated by increasing AI-based automation of scientific research and the importance and difficulty of lab research. The key contributions are a biology lab simulation environment, including implementations of simulated phenomena that are crucial for lab biology research; a benchmark of biological tasks; and systematic evaluation of VLAs and simpler imitation learning baselines.

The paper discusses the simulation implementation: first, a process for digitizing assets, particularly lab equipment. Next, physics implementations to augment existing simulation capability: thread mechanism, detent mechanism, eccentric mechanism, and quasi-static liquid. For rendering, the benchmark uses basic rendering from MuJoCo, contributes advanced rendering blending MuJoCo's simulation state with blender's rendering pipeline, and contributes a reactive user interface. The benchmark itself includes randomized scene initialization, procedural demo generation, and task evaluation via predefined status checks on various pieces of state information. Tasks come in three difficulty levels, increasing in difficulty of vision and manipulation precision as well as task requirement/language instruction variation and abstraction/complexity.

The experiments are conducted by fine-tuning VLAs (pi_0, pi_0.5, and RDT) on either 100 or 20 demo trajectories. Non-VLA imitation learning baselines are trained from scratch on the data. Results show that RDT is more consistent on easy tasks, while pi_0 and pi_0.5 are better in challenging scenarios. No significant advantage is seen from pi_0.5 over pi_0 on medium- to hard-level tasks, which the paper ascribes to high-level complexity and reasoning requirements. The paper also presents failure analysis: failures on easy tasks are usually due to gripper slippage, whereas medium and hard tasks are due to compounding precision manipulation issues; the paper suggests that this indicates a need for algorithms with stronger closed-loop learning abilities, including RL. Failures on these tasks are also due to language understanding limitations, and partial observability. These are exacerbated by low input resolutions, a limitation of the benchmark. Imitation learning baselines are able to match performance on a single task when they are trained on that task alone, but not in the multi-task setting.

**Strengths:**

### Quality
- Design makes sense and is well-explained.
- Task difficulty clearly matters and is differentiating, which is key - I find this to be the hallmark of a good benchmark
- Failure analysis is really promising (see weaknesses for thoughts on how to improve, but great addition)
- Experiment suite is very good! I especially appreciate the holdout experiments, both because I feel it's important to fundamental ML science, and because I suspect this is a domain where there will be plenty of unseen info for a long time.
- Simulation additions are clearly valuable and significant
### Clarity
- Incredibly well-written paper overall! Well-structured and very easy to follow.
- Visual figures are really useful
### Originality and significance
- From what I understand, there are bio-lab benchmarks, but this one's design is unique and well-proven
- Very important problem and this paper presents a benchmark that took considerable work in various aspects, so no concerns about significance.

**Weaknesses:**

### Quality
- Some simulation features presented do already exist, though I suspect not with the fidelity/design choices needed here, meaning they do not take away from the contributions of this paper. However, they would be worth adding to RW: transparency and liquids are simulated in multiple simulators including OmniGibson (BEHAVIOR-1K, Li et al. 2022), and a threading mechanism is used in TRANSIC (Jiang et al., 2024).
- Failure analysis needs more detail. Would benefit from examples of failure modes, to convince the reader that the explanations are correct
- Rendering issues leading to low-quality visual observations is a failure of the benchmark, not the method, and should be attributed accordingly
- Some analysis/stats on the quality of the simulation would be valuable, though I think it's less central than better exposure of the method results
### Clarity
- Results are presented in simple tables. This makes them very hard to follow, especially for a benchmark where all the statistics are important. It would really help to have charts. This should include direct comparison between the baselines and VLAs, comparison between similar tasks, and direct comparison between difficulty levels, as in my opinion this is the most salient comparison toward judging the quality of the benchmark
- Overall, results section just needs more structure. The figures I talked about are a big part of it, but also a claims-driven structure - right now it's quite an info dump, which is hard to follow.

The reason for the 6 is mainly that I want the results to be presented better. Given that, I would recommend a higher score.

**Questions:**

- What is the difference between "what" and "why" in Fig 1?
- Questions about grounding failure analysis, as listed in Weaknesses
- Where do the procedural policies for demo generation come from?

---

> ### Author Response · Authors · 2025-11-19
> **Response (1/2)**
>
> We thank the reviewer for their valuable comments, which have helped us improve the manuscript. Our detailed responses are provided in the subsequent sections.
>
> **W1** (prior simulation features):
> We thank the reviewer for pointing out these highly relevant works. We have revised the Related Work & Section 3.2 to include a discussion of these simulation features.
>
> **W2 & Q2** (failure mode analysis):
>
> We appreciate the reviewer's suggestion. We have conducted trajectory-level failure analyses during model evaluation, and we will include it in upcoming paper revision. Broadly, the observed failures are highly interpretable and fall into three main categories:
>
> 1. **Insufficient precision in fine manipulation**: Inserting a centrifuge tube to a rack or rotor slot require tight spatial tolerances. VLAs often exhibit drift or inconsistent end-effector alignment, leading to tubes missing slots, bouncing off edges, or tipping over.
> 2. **Lack of adaptiveness and recovery behavior**: For operations requiring continuous feedback—e.g., threading motions in Screw/Unscrew Cap—the policies frequently commit to an open-loop trajectory and do not adjust when contact deviates from the demonstration. We found no evidence that robust recovery strategies emerge from pure imitation; this suggests that either online adaptation or additional modalities (e.g., force/torque sensing) may be necessary.
> 4. **Inaccurate correlation across modality**: In tasks involving visual state understanding and instruction following (e.g., "Transfer Centrifuge Tube" or "Operate Thermal Mixer Panel"), models sometimes confuse similar-looking states or average across visually ambiguous features. This leads to actions that are semantically plausible but contextually wrong—such as pressing the wrong interface region. These errors highlight limitations in current VLA pretraining for precise visio-linguistic grounding in scientific environments.
>
> **W3** (visual quality at small scale):
>
> Thank you for pointing this out. We agree it is important to clearly separate limitations of the benchmark from limitations of the models.
>
> To clarify: the benchmark itself does not impose a low-quality rendering constraint. The AutoBio simulator and renderer can generate high-fidelity images at arbitrary resolutions. In principle, our tasks can be evaluated with significantly higher visual resolutions.
>
> The bottleneck arises instead from current VLA pipelines, which uniformly downsample all images to 224x224 to match their ViT and VLA pretraining configuration. Changing this resolution only at fine-tuning time would break compatibility with pretrained encoders and introduce training–inference mismatches. Moreover, increasing resolution substantially increases the number of visual tokens (e.g., 2× resolution → ~4× tokens), which results in more than a 4× cost increase in both training and inference. These constraints—not the benchmark—are what limit usable visual resolution in our experiments.
>
> We will revise the paper to clarify this distinction so that the rendering-related performance degradation is correctly attributed to model-side preprocessing and architectural constraints rather than to deficiencies in AutoBio.
>
> **W4** (simulation performance):
> We here provide a brief analysis for simulation speed of AutoBio. Our simulation setup is as follows. Each task is simulated with a base time step of **2ms**, and control inputs are updated every **10 steps** (i.e., **50Hz** control frequency). The simulation time can be decomposed into two main components: **physics simulation** and **rendering**.
> - **Physics Simulation**: Each physics step takes between **20μs and 750μs**, depending on the task complexity. Consequently, simulating one action chunk (500 steps) takes **10ms to 375ms**.
> - **Rendering**: Each action chunk requires rendering a bundle of **2–3 images**. **MuJoCo** rendering takes **100–200μs** per image which is negligible, while **Blender** takes **50–100ms**.
>
> The performance measurements reported above are obtained on a workstation with **Intel(R) Core(TM) i9-13900K** and **NVIDIA GeForce RTX 4090**.

---

> ### Author Response · Authors · 2025-11-19
> **Response (2/2)**
>
> **W5 & W6** (result presentation clarity):
>
> Thank you for this helpful suggestion. We agree that the results section in a benchmark paper must present comparisons in a way that is easy to interpret across multiple axes (model class, task type, and difficulty level). As you note, dense tables alone make it difficult to see the larger patterns, especially when no single method dominates.
>
> In response, we have substantially reorganized our results section. We replaced the two large tables with a single grouped stacked bar chart that directly compares VLAs and imitation-learning baselines across all tasks. The original detailed tables are now in the appendix for readers who want fine-grained statistics. We also restructured the accompanying discussion to be claims-driven rather than enumerative, so that the narrative highlights the key takeaways.
>
> We believe these changes improve readability and make the benchmark's insights more accessible.
>
> **Q1** (Fig 1 clarification):
>
> The difference lies in the level of abstraction. In Figure 1, the "What" column illustrates high-level biological experimental operations commonly perform in laboratories. In contrast, the "Why" column demonstrates how these complex operations can be decomposed into a series of fundamental, atomic manipulation skills. For instance, the operation of "Centrifugation" involves more basic tasks like sample transfer (loading tubes into the rotor) and operating the centrifuge machine. This taxonomy of atomic skills provide the methodological foundation and justification (the "why") for designing our simulator and the benchmark tasks.
>
> **Q3** (Demonstration synthesis):
> The procedural (expert) policies for demo generation are derived from standard operating procedures (SOPs) for the respective laboratory techniques. Specifically, for each task, we manually decompose the operational workflow into sequential subtasks (for example, reach-grasp-lift in the "Pick up centrifuge tube" task). For each subtask, we define a programmed end-effector motion path based on the subtask initial position and annotated keypoints relative to target objects, then compute the corresponding robot joint trajectory using inverse kinematics combined with time-optimal path parameterization (TOPP). Subtasks are concatenated and executed via PD control.

---

### Official Review · Reviewer_CmrM · 2025-11-01

**Soundness:** 4
**Presentation:** 4
**Contribution:** 4
**Rating:** 8
**Confidence:** 4

**Summary:**

The authors propose a high-fidelity simulator targeting the biology lab automation domain for robotics. This domain has particular characteristics, like the preponderance of specialized machines (e.g. thermal mixers, centrifuges), clear / colored liquids, and transparent/translucent material (e.g. test tubes) which present difficulties for off-the-shelf robotic simulators like MuJoCo and its base OpenGL renderer. To solve this, they introduce a pipeline which digitizes real lab instruments through a 3D Gaussian Splat representation, adds  MuJoCo plugins for specialized physics (e.g. thread, detent, and eccentric mechanisms, liquid deformations) and more realistic rendering for transparent liquids through Blender physically-based rendering (PBR) and dynamically loading texture maps to make simulated machine control panels and displays dynamic.

In addition, given this simulator, the authors then develop a benchmark targeting 16 biologically-grounded tasks across three difficulty levels (easy, medium, hard) to systematically evaluate language-guided robotic manipulation in lab protocols. They then build infrastructure to generate 100 demonstration trajectories and then carefully evaluate state-of-the-art VLAs and smaller imitation learning baselines after fine-tuning them on the demonstrations.

**Strengths:**

Overall, the paper provides clear motivation for a focused effort on targeting simulation for the biology lab use-case, a significant area of both research and industrial importance and therefore has the potential for high impact.

The work is a high-quality, well-executed set of improvements targeting the precise difficulties of simulation in this domain. In particular, the improvements to base Mujoco (both physics and rendering) are extremely relevant to the domain. In particular, interacting with the control panels / displays of specialized machines is relatively rare in robotic simulation, and dynamically loading texture maps to achieve dynamic feedback is an example of a small but highly important infrastructural addition.

The primary originality of the work lies in the comprehensiveness and thoughtfulness of new techniques targeting the domain itself. While each technique (e.g. adding fluid modeling) is not extremely novel, the breadth of these which target domain-specific simulation issues demonstrates creativity.

Finally, the careful set of benchmark tasks and baseline evaluation of VLAs / imitation learning policies is likely to be highly impactful, demonstrating the clear gap between current SOTA methods on this benchmark. The analysis of these results is thorough.

**Weaknesses:**

Especially given that most of the work is aimed at achieving stronger realism, the paper would be strongly improved by any real-world experiments demonstrating that the methods result in transfer onto real robotic hardware.

A significant aspect of the domain is executing longer-term procedures, and much of the work is motivated by the long task horizons. However, the main text has minimal emphasis on this aspect in the experiments / tasks, and the long-horizon task in the appendix is simply concatenation of short trajectories.

The simulation is also focused on static manipulation, which may preclude true autonomous execution of multiple steps in more realistic lab environments given the lack of mobility.

The evaluation trained separate models for each task and evaluated tasks independently, while VLAs are motivated by cross-task transfer and language generalization; evaluation of a VLA trained and tested across multiple tasks could be informative.

**Questions:**

As you mention, low input resolutions exacerbate instruction following. Is this an issue with the task setup in the "Operate thermal mixer panel" which actually prevents the task from being fully executed?

Can you expand on the physical realism of the "aspirate with pipette" task? Presumably some level of tactile or force feedback is necessary for working with pipettes in the real world in order to control the amount of liquid aspirated.

Is there any value in providing instrument readings as an observation to the policies directly? Presumably some lab devices are IoT-enabled, and it could be a reasonable test of whether perception is fundamentally limiting policy performance or not.

Do you expect increasing the number of demonstrations to significantly affect performance on the medium/hard tasks? Further investigation on data scaling effects (perhaps for a subset of the tasks) could be highly informative.

---

> ### Author Response · Authors · 2025-11-19
> **Response (1/2)**
>
> We are grateful to the reviewer for their constructive and thoughtful feedback, which have helped us to refine our work. Our responses to each comment follow.
>
> **W1** (real-world experiment):
> We thank the reviewer for highlighting the importance of sim2real transfer evaluation in assessing simulation benchmarks. We fully agree that demonstrating alignment between simulation and real-world performance is an important long-term goal for simulation environments, particularly those intended to support policy learning. However, conducting real-world robotic experiments in a biology lab setting presents unique challenges. Many of the instruments involved are delicate, costly, and require careful handling. Given our current infrastructure, we chose to prioritize simulator fidelity and safety for early-stage development. We plan to revisit sim2real validation as our hardware and automation infrastructure mature.
>
> **W2** (long-horizon):
>
> We thank the reviewer for this constructive point. Long-horizon biology protocols are indeed a central motivation for AutoBio. In the current submission, we chose to focus our main experiments on shorter horizon tasks because our demonstration-synthesis infrastructure is not yet mature enough to robustly support flexible, diverse, and error-tolerant long-horizon data generation.
>
> We fully agree that more long-horizon tasks would strengthen the benchmark, and this is a major direction of our ongoing work. We are actively developing: (1) automatic scene composition to facilitate multi-stage protocols, (2) standardized representations for multi-step experimental procedures, and (3) LLM-driven synthesis pipelines that can generate coherent long-horizon demonstrations.
>
> We also note that long-horizon execution presents challenges beyond data generation—for instance, controlled recovery from intermediate failures is unlikely to emerge from large-scale imitation of only successful trajectories. These issues remain open problems in VLA research.
>
> **W3** (chassis mobility):
>
> Thank you for this insightful comment. We agree that mobility is a crucial capability for enabling multi-step autonomous operations in realistic laboratory settings. In response, we have incorporated a mobile dual-arm platform (with an Agilex Ranger Mini base) into our simulation assets. Moving forward, we will develop more mobile manipulation tasks based on this robot system. This mobility will serve as the foundation for executing long-horizon tasks, where the mobile base will act as a link between discrete experimental procedures.
>
> **W4** (multi-task):
>
> Our paper includes one configuration that is partially multi-task: in Appendix "Additional Experiment Results for Long-Horizon Task", we train a VLA model on a mixture of demonstrations from two subtasks. This experiment shows that (1) the VLA achieves reasonable per-task performance when trained jointly on those tasks, but (2) such mixed-task training does not enable the model to automatically compose or concatenate sub-trajectories to solve the corresponding long-horizon task. This highlights a gap that AutoBio is designed to surface.
>
> Regarding broader multi-task evaluations: our demonstration-generation and training pipelines do not impose restrictions on multi-task training. However, current open-source VLAs are typically fine-tuned for a single robotic embodiment—their action heads are neither embodiment-agnostic nor explicitly embodiment-conditioned—so jointly training tasks collected from different robots would be technically inconsistent. For this reason, we did not mix tasks across embodiments in the main paper.
>
> That said, we agree that a stronger multi-task baseline is valuable. In the upcoming revision, we will add an experiment that jointly trains on all uni-manual UR5e tasks to provide a more complete assessment of multi-task performance in a consistent embodiment setting.

---

> ### Author Response · Authors · 2025-11-19
> **Response (2/2)**
>
> **Q1** (UI resolution):
>
> Thank you for raising this point. Low input resolution is indeed a contributing factor to the performance drop on "Operate Thermal Mixer Panel". At 224x224, important UI elements lose fidelity, which makes precise button targeting and reading small text more difficult. This compounds the underlying semantic challenge of the task: the model must ground fine-grained linguistic instructions like "set speed to 500 rpm" to specific visual interface elements. This cross-modal alignment is a core research challenge, and the low-resolution input further constrains the model's ability to resolve the necessary visual details for robust performance.
>
> Importantly, this limitation does not come from the AutoBio simulator or task design. Our rendering pipeline supports arbitrarily high-resolution observations, and the benchmark could operate at higher fidelity. The constraint arises from the models' preprocessing pipelines, which uniformly downsample all inputs to 224×224 to match their ViT and VLA pretraining setup. Changing this unilaterally in our fine-tuning would create training–inference mismatches and invalidate the pretrained backbones; moreover, doubling image resolution would approximately quadruple the number of visual tokens, greatly increasing training and inference cost.
>
> **Q2** (Tactile feedback):
>
> In our current "aspirate with pipette" task, the pipette is modeled as a adjustable-volume mechanical pipette. This pipette can aspirates the exact pre-set volume of liquid by depressing the plunger to its mechanical stop (a fixed displacement) and then releasing it. This allows us to model successful aspiration based on position control alone without the need for force feedback. We acknowledge that force feedback becomes crucial for tasks involving rich contacts, such as screwing caps. Integrating force/tactile sensing is a key direction for our future work to enhance the physical realism.
>
> **Q3** (Direct sensor reading observation):
>
> We agree that providing instrument readings (e.g., numerical sensor values or device states) as observations can be valuable. Such signals offer low-noise ground truth and would help disentangle whether failures stem from perception limits or from higher-level reasoning. AutoBio can already expose these states with minimal configuration; the simulator itself does not restrict such access.
>
> That said, our focus in the paper is to evaluate VLAs under their intended usage paradigm—a unified, vision-centric interface that mirrors human sensing, where the model must infer instrument state directly from the visual scene. Incorporating device-specific structured data raises a modeling question that current VLA architectures are not designed for: they lack a principled way to fuse heterogeneous numeric or symbolic sensor channels alongside vision and language. Doing so would require either (1) developing new multimodal architectures that explicitly incorporate structured device-state inputs, or (2) serializing these signals into language tokens, a direction we agree is promising but remains underexplored.
>
> **Q4** (Data scaling):
>
> Scaling up the demonstrations is indeed quite effective to address some failure patterns. Specifically regarding the "Operate thermal mixer panel" task, the robot could execute seemingly plausible trajectory, but the quantitative results after adjustment is rarely accurate, indicating insufficient connection between language (the setpoint), vision (current panel state) and proprioception-action. Scaing up to **1000** demos could boost pi0/pi05's percentage score from 10 to 50, while further scaling shows no obvious benefits. Since we did not conduct this scaling experiement on all tasks, we would include this preliminary result in appendix in upcoming paper revision.

---

### Official Review · Reviewer_qBEk · 2025-11-01

**Soundness:** 4
**Presentation:** 3
**Contribution:** 3
**Rating:** 8
**Confidence:** 3

**Summary:**

The paper proposes AutoBio, a simulation benchmark designed specifically for laboratory tasks and environments. The authors provide a laboratory equipment asset generation pipeline, relevant physics plugins, rendering which supports e.g. transparent materials, and a data generation pipeline. The authors also benchmark VLA and IL baselines on tasks of varying levels of difficulty.

**Strengths:**

- Useful task suite in an area of robotic manipulation with few realistic benchmarks
- Careful consideration of physics, rendering, assets, etc in the context of biology tasks
- VLA and IL baselines are relevant and highlight weaknesses in more complex tasks
- The presentation is clear and contributions well-explained

**Weaknesses:**

- Seeing as the realistic assets, physics, and rendering are a central focus, validation on a real robot setup (even on the simpler tasks) would support claims of realism
- The paper notes VLAs may perform well as multi-task agents in the discussion, however this setting is not evaluated

**Questions:**

- Are benchmarks on simulation performance available (e.g. simulation speed) to gauge evaluation speed, and potentially the applicability of online learning methods?
- The authors note the potential multitask capability of the VLA models. Did authors run multitask experiments on the VLA and IL models?
- Are there future plans to provide additional tasks (e.g. longer horizon, more subtasks, etc), leveraging the same physics and rendering framework?

---

> ### Author Response · Authors · 2025-11-19
>
> Response:
>
> We thank the reviewer for their time and insightful comments, which have helped us significantly improve the quality and clarity of this manuscript. After carefully considering all the feedback, our point-by-point responses to the specific comments are detailed below.
>
> **W1** (real-world experiment):
>
> We thank the reviewer for highlighting the importance of sim2real transfer evaluation in assessing simulation benchmarks. We fully agree that demonstrating alignment between simulation and real-world performance is an important long-term goal for simulation environments, particularly those intended to support policy learning. However, conducting real-world robotic experiments in a biology lab setting presents unique challenges. Many of the instruments involved are delicate, costly, and require careful handling. Given our current infrastructure, we chose to prioritize simulator fidelity and safety for early-stage development. We plan to revisit sim2real validation as our hardware and automation infrastructure mature.
>
> **W2 & Q2** (multi-task):
>
> Our paper includes one configuration that is partially multi-task: in Appendix "Additional Experiment Results for Long-Horizon Task", we train a VLA model on a mixture of demonstrations from two subtasks. This experiment shows that (1) the VLA achieves reasonable per-task performance when trained jointly on those tasks, but (2) such mixed-task training does not enable the model to automatically compose or concatenate sub-trajectories to solve the corresponding long-horizon task. This highlights a gap that AutoBio is designed to reveal.
>
> Regarding broader multi-task evaluations: our demonstration-generation and training pipelines do not impose restrictions on multi-task training. However, current open-source VLAs are typically fine-tuned for a single robotic embodiment—their action heads are neither embodiment-agnostic nor explicitly embodiment-conditioned—so jointly training tasks collected from different robots would be technically inconsistent. For this reason, we did not mix tasks across embodiments in the main paper.
>
> That said, we agree that a stronger multi-task baseline is valuable. In the upcoming revision, we will add an experiment that jointly trains on all uni-manual UR5e tasks to provide a more complete assessment of multi-task performance in a consistent embodiment setting.
>
> **Q1** (simulation performance):
>
> Before addressing simulation speed, we first clarify our simulation setup. Each task is simulated with a base time step of **2ms**, and control inputs are updated every **10 steps** (i.e., **50Hz** control frequency). The simulation time can be decomposed into two main components: **physics simulation** and **rendering**.
> - **Physics Simulation**: Each physics step takes between **20μs and 750μs**, depending on the task complexity. Consequently, simulating one action chunk (500 steps) takes **10ms to 375ms**.
> - **Rendering**: Each action chunk requires rendering a bundle of **2–3 images**. **MuJoCo** rendering takes **100–200μs** per image which is negligible, while **Blender** takes **50–100ms**.
>
> The performance measurements reported above are obtained on a workstation with **Intel(R) Core(TM) i9-13900K** and **NVIDIA GeForce RTX 4090**.
>
> While we have not included online learning methods in this study, our tasks can be wrapped as Gymnasium environments and are compatible with RL approaches. We are actively investigating RL-based fine-tuning of the VLA model, which we plan to include in upcoming iterations of this work.
>
> **Q3** (task expansion):
>
> Yes, providing more tasks, including long-horizon tasks and manipulation tasks using dexterous hands, is an important future direction for this project.

---

### Author Response · Authors · 2025-11-21
**General Response on Multi-Task Training With Extra Experiment**

Several reviewers raised questions regarding the multi-task training setting in AutoBio.

First, we clarify that the AutoBio demonstration-generation pipeline and training infrastructure do not impose limitations on multi-task learning. In principle, any subset of tasks can be mixed during training. The practical constraint comes from current open-source VLA architectures, which are typically fine-tuned for a single robotic embodiment and use action heads that are neither embodiment-agnostic nor explicitly embodiment-conditioned. Mixing tasks collected from different robots would therefore yield incompatible action spaces. For this reason, the main paper reports only single-task fine-tuning.

To directly address reviewers’ concerns, we have now added a multi-task experiment that jointly trains on all uni-manual UR5e tasks, using 100 demonstrations per task. This provides a clean evaluation of multi-task performance within a consistent embodiment.

The setting and preliminary results are shown below:

| Task \ Algorithm | $\pi_0$ | $\pi_{0.5}$ |
| -------- | -------- | -------- |
|  Close thermal cycler lid    | 0.85     | 0.86     |
|  Open thermal cycler lid    | 0     | 0.06     |
|  Transfer centrifuge tube    | 0     | 0.01     |
|  Operate thermal mixer panel    | 0     | 0     |
|  Load centrifuge rotor    | 0     | 0     |
|  Close centrifuge mini lid    | 0.98    | 0.95     |
|  Close centrifuge 5430 lid    | 0.01     | 0.03     |
|  Close centrifuge 5910 lid    | 0.65     | 0.77     |

Relative to the single-task results in the main paper, multi-task fine-tuning produces task-dependent degradation. We visually inspected all task rollouts and verified that policies do condition appropriately on each task's context (visual input & prompt); failures arise only after many visually correct steps.

Across tasks, the dominant failure mode is loss of manipulation precision. Small trajectory deviations—e.g., slight misalignments when reaching for levers or lids—lead to collisions or unstable grasps ("Open thermal cycler lid", "Close centrifuge 5430 lid"), whereas such errors do not occur under single-task training. This suggests that achieving sample-efficient multi-task fine-tuning in manipulation requires joint progress in both VLA architectural design and data-collection strategies.

We will integrate this experiment and discussion into the revised version of the paper.

---

### Meta-Review · Area_Chair_6A1f · 2025-12-22

**Summary:**

The submission introduces a simulation framework and benchmark designed to evaluate robotic automation in biology laboratory environments.  Reviewers liked the idea but were concerned about the realism and sim-to-real gap, as well as the evaluation and presentation.

**Reviewer Concerns:**

Unfortunately, most concerns appear to be unaddressed by the rebuttal.

**Reviewer Scores:**

Reviewers will most likely maintain their scores of 8, 8, 6, 6.  Overall, the strengths of the submission may outweigh its weaknesses.

---

### Decision · Program_Chairs · 2026-01-26

Accept (Poster)